# Surface Environment and Energy Density Effects on the Detection and Disinfection of Microorganisms Using a Portable Instrument

**DOI:** 10.3390/s23042135

**Published:** 2023-02-14

**Authors:** Sungho Shin, Brianna Dowden, Iyll-Joon Doh, Bartek Rajwa, Euiwon Bae, J. Paul Robinson

**Affiliations:** 1Department of Basic Medical Sciences, Purdue University, West Lafayette, IN 47907, USA; 2Bindley Bioscience Center, Purdue University, West Lafayette, IN 47907, USA; 3School of Mechanical Engineering, Purdue University, West Lafayette, IN 47907, USA; 4Weldon School of Biomedical Engineering, Purdue University, West Lafayette, IN 47907, USA

**Keywords:** contamination detection, disinfection, energy density, microorganisms, portable device

## Abstract

Real-time detection and disinfection of foodborne pathogens are important for preventing foodborne outbreaks and for maintaining a safe environment for consumers. There are numerous methods for the disinfection of hazardous organisms, including heat treatment, chemical reaction, filtration, and irradiation. This report evaluated a portable instrument to validate its simultaneous detection and disinfection capability in typical laboratory situations. In this challenging study, three gram-negative and two gram-positive microorganisms were used. For the detection of contamination, inoculations of various concentrations were dispensed on three different surface types to estimate the performance for minimum-detectable cell concentration. Inoculations higher than 10^3^~10^4^ CFU/mm^2^ and 0.15 mm of detectable contaminant size were estimated to generate a sufficient level of fluorescence signal. The evaluation of disinfection efficacy was conducted on three distinct types of surfaces, with the energy density of UVC light (275-nm) ranging from 4.5 to 22.5 mJ/cm^2^ and the exposure time varying from 1 to 5 s. The study determined the optimal energy dose for each of the microorganisms species. In addition, surface characteristics may also be an important factor that results in different inactivation efficacy. These results demonstrate that the proposed portable device could serve as an in-field detection and disinfection unit in various environments, and provide a more efficient and user-friendly way of performing disinfection on large surface areas.

## 1. Introduction

Food safety is an ongoing challenge for national food supply chains and international trade [1,2,3]. To tackle this problem issue, microbiological food-safety research is categorized into two major areas covering detection and intervention. Detection aims to find the pathogens and provides information and direction for the downstream process of intervention. This can range from traditional culturing (gold standard) to more recent polymerase chain reaction (PCR) methods [4,5,6], biochemical methods [7], and novel instrumentations such as the elastic mass spectrometry (MALDI-TOF) light-scattering (ELS) technology [8,9,10] and laser-induced breakdown spectroscopy (LIBS) [11], as well as others. Most current detection methods rely on sampling-based techniques that require a significant length of time-to-result. Intervention utilizes results from the detection step and provides solutions to maintain a safe environment for growers, food manufacturers, and consumers. Hazard analysis and critical control point (HACCP) is a widely accepted food-safety management system by many stakeholders in the food industry [12]. One of the key points of HACCP is applying disinfection steps to ensure that biological hazards are removed.

Disinfection methods are classified as biological, physical, or chemical. Biological methods utilize biological agents to kill or inactivate the target pathogen or toxin [13,14], while physical intervention includes irradiation and filtering [15,16]. Chemical disinfection agents, most notably chlorine or ozone, are widely used in the food industry to disinfect surfaces [17]. While there are advantages to using chemical methods to effectively disinfect large surface areas (low cost in particular), the associated risk, such as the hazardous nature of many chemicals, has created opportunities for alternative methods of intervention; for instance, UV irradiation has been widely used to disinfect large surfaces in clinical settings. Various optical techniques aimed at pathogenic bacteria and viruses have also been documented [18,19]. Hospitals are at the frontier of adopting optical disinfection methods, often based on autonomous systems [20,21,22]. While effective, these robotic devices used in hospitals require considerable capital investments.

The recent development of portable disinfection instruments has brought cost-effective and efficacious technology [23,24,25,26,27]. Some systems focus on the disinfection of individual items within a container or room, while others are designed to irradiate large areas of interest with multiple UV bands. Ideally, these systems should provide disinfection capability with multiple bands while maintaining portability [28]. A recent report utilizing the CSI-D+ device (also used in this study) demonstrated simultaneous fluorescence-based detection and UV-based sterilization capability in a single battery-operated unit. Such a design could potentially allow inspectors to employ only one unit for both assessment and disinfection. However, the effect of various confounding factors, such as the type of a substrate or an effective energy dose, are still unclear, and only a few publications thoroughly investigate the disinfection process using handheld instruments [23,29]. Compared to Sueker et al. [28], this work focuses on expanding the applicability of the proposed instrument in surface contamination detection and disinfection. This was accomplished by preparing samples containing multiple types of microorganisms and utilizing multiple types of surfaces that may be found in food processing facilities or hospitals.

Here, we report a validation study describing the detection and disinfection efficacy of the CSI-D+ device for both gram-negative and gram-positive organisms. The limit of detection size and effective energy doses have been estimated with multiple surfaces (liquid, semi-solid, and solid). These data will guide the actual implementation of this technology and the innovative handheld instrument.

## 2. Materials and Methods

### 2.1. Sample Preparation

Three different gram-negative organisms, *Escherichia coli* (ATCC 25922), *Klebsiella pneumoniae* (ATCC 13883), and *Salmonella enteritidis* (ATCC 13076), and two different gram-positive organisms, *Listeria innocua* (ATCC 33090) and *Staphylococcus aureus* (ATCC 23235), were used in this study. All organisms were cultured at 37 °C after plating on Trypticase soy agar (TSA). A single colony from each plate was diluted in sterile phosphate-buffered saline (PBS) over a 4-log range in sterile glass tubes before use. The initial starting dilution was approximately 10^7^ CFU/mL and estimated through colony counting.

### 2.2. Portable Disinfection Unit

A commercially available instrument (CSI-D+; SafetySpect Inc., Grand Forks, ND, USA) was selected as test equipment for disinfection of microbiological samples. This instrument included 405 nm (violet) as a source for detection mode and 275 nm (UVC) wavelength LED arrays for the exposure during both detection and disinfection mode, and a light detection and ranging (LiDAR) module to determine the surface distance, as shown in Figure 1a. This device includes an RGB, a UV camera for image acquisition, and a display screen, as shown in Figure 1b. Appropriate personal protective equipment (PPE), such as safety goggles, was mandatory during all disinfection procedures.

### 2.3. Detection Procedures

Three different environments were selected to test the CSI-D+’s ability to detect microorganisms: liquid samples, semi-solid agar surface, and ceramic tile surface. All images were captured at a fixed working distance of 12.7 cm, and the excitation source and power conditions were chosen to be 405 nm and 40 mW. For liquid samples, 10 μL aliquots of inoculation duplicates were placed on an eight-well glass slide (well aperture 8 mm). Three different concentrations (10^8^, 10^7^, and 10^6^ CFU/mL) were diluted using sterile MilliQ water since PBS has auto-fluorescence at 405 nm. Next, 10 μL of organisms (10^3^ CFU/mL dilution) was spread-plated onto TSA plates using an L-shaped spreader. Each measurement was conducted while increasing incubation time every three hours from 0 to 12 h at 37 °C. Lastly, 100 μL inoculation solution was spread on the surface of a black ceramic tile (Dal-tile Corporate, Dallas, TX, USA) in a 5-cm circle diameter. The images for both the glass slide and the ceramic tile were measured after 1 h of drying in a class-II laminar hood.

### 2.4. Disinfection Procedures

Three different growth environments were selected for comparison, as shown in Figure 2. The bacteria were suspended in PBS and thoroughly mixed to ensure equal exposure, and their absorbance in the UV range was negligible [30]. In detail, 100 μL of the varying concentrations of each organism were dispensed into individual wells of a 96-well plate in triplicate. The specific area (4 × 3 arrays of an individual well for every energy density) was subsequently exposed to UVC light while increasing energy density from 4.5 to 22.5 mJ/cm^2^ with a fixed working distance of 23 cm. Note that an irradiance of 4.5 mW/cm^2^ was measured at a fixed working distance, and the energy density was estimated multiplying the irradiance by the exposure time of UVC. After UVC exposure, 10 μL was removed from each well and spread onto TSA plates using an L-shaped spreader [31]. All plates were incubated at 37 °C for 24 h, followed by colony counting.

For the second setting, TSA plates were chosen, and 10 μL of each organism was removed from each dilution and spread-plated onto TSA plates in triplicate. The entire area of the TSA plate (100 mm diameter) was subsequently exposed to UVC light and directly compared with the control plates with no UVC exposure. The UVC energy density and working distance were the same as in the liquid-sample testing. Note that all plates were positioned in the center of the exposure area, and the edge of the plate was exposed with a uniform UVC dose (>85%) estimated by a power meter. All plates, including both exposed plates and unexposed plates (control), were incubated and counted.

Third, a ceramic tile was chosen as a solid surface environment since these tiles are common surfaces in kitchens and hospitals [32,33]. A bright white ceramic wall tile (Dal-tile Corporate, Dallas, TX, USA) was obtained from a local home-improvement store. Tiles measuring 108 mm × 108 mm × 8 mm were selected to match the exposure area of the TSA plates. After all tiles were autoclaved, 100 μL of each organism was spread on the top surface of the tile using an L-shaped spreader. The whole area of the tile was subsequently exposed to UVC light, maintaining the same exposure conditions as in the first and second cases. Organisms on the tile surface were then swabbed from top to bottom with a sterile cotton swab dipped into sterile PBS, followed by swabbing onto TSA plates. This procedure was repeated three times using a different swab, each time in a different location on one tile surface where the bacteria were spread. A total of three TSA plates from one tile were incubated at 37 °C for 24 h for counting. The control set was handled in the same manner except for UVC exposure.

A total of 360 PBS samples, 90 agar dishes, and 90 tile plates were tested to count. Note that only one fixed diluted concentration (10^3^ CFU/mL from 10^4^ dilutions) was selected in both TSA plates and the measurement of the tile after various concentration tests in the liquid matrix. The killing rate and log reduction were calculated according to Equation (1) [16] and Equation (2) [29,34], respectively. *N* is the number of colonies after exposure, and *N*_0_ is the initial number of colonies from control plates.
(1)$$\mathrm{Killing}\;\mathrm{rate}=100-\left(\frac{N}{N_{0}}\times 100\right),$$
and
(2)$$\mathrm{Log}\;\mathrm{reduction}=\log_{10}\left(\frac{N}{N_{0}}\right).$$

## 3. Results

### 3.1. Contamination Detection

Contamination detection results are shown in Figure 3. Each image represents a serially diluted liquid drop on (a) the slide glass, (b) the TSA plate with spreading, and (c) the black tile surface with spreading via sterile swabs. The top row shows raw images from the CSI-D+ device, and the bottom row includes the processed images. Processed images were converted into grayscale bitmaps, and ROI extractions were performed in Matlab. A 2D median filter and binary image conversion (luminance threshold = 0.4) were sequentially applied to isolate the surface contamination. It was shown that inoculation of 4.51 × 10^4^ (dispensing in slide glass) and 3.18 × 10^3^ (spreading on tile surface) CFU/mm^2^ concentration (average of two determinations) is the current limit of detection after dispensing or spreading liquid samples. In addition, a single colony size greater than 0.15 mm could be detected, which takes approximately 8 h of incubation at 37 °C on a TSA plate.

### 3.2. Disinfection in the Liquid Sample

The disinfection results for the liquid samples are shown in Figure 4. The killing rates were calculated for the five different microorganisms. The concentrations shown in the left and right columns were 10^4^ and 10^3^ CFU/mL, respectively. The results from higher concentrations (10^6^ and 10^5^ CFU/mL) are shown in the Appendix A. The solid black symbols indicate the killing rate values. The whiskers show standard deviations calculated from triplicates. A sigmoidal–logistic fitting curve is shown as a red dashed line.

Figure 4 shows that the effective dose for disinfection varied depending on the type of bacterial species. First, *K. pneumoniae* colonies in various concentrations (10^6^~10^3^ CFU/mL) were completely removed by a dose of at least 4.5 mJ/cm^2^ (exposure time of 1 s). Second, it was shown that a 10.0 mJ/cm^2^ (exposure time of 2 s) effective dose could eliminate three species (*S. enteritidis*, *L. innocua*, and *S. aureus*). Third, *E. coli* had a relatively higher resistance to UVC than other species. As a result, an effective energy dose above 20.0 mJ/cm^2^ (exposure time of 4 s) was required for a bactericidal effect. In addition, it was shown that the influence of concentration on minimum effective dose was negligible in this energy density range (4.5–22.5 mJ/cm^2^). Only obvious improvement appeared at lower energy density, such as in Figure 4e at 4.5 mJ/cm^2^. The similar effective doses were also visually represented in Appendix A.

### 3.3. Disinfection in Agar Media

The disinfection results in nutrient-rich agar media (TSA media, red circle) and liquid sample (PBS, black rectangle) are shown in Figure 5. The killing rates are represented along with energy density increment, and the measurements were conducted at fixed concentrations (10^3^ CFU/mL). It was shown that similar killing rates were achieved in each species and energy densities. Similarly, each effective energy dose used for *E. coli*, *K. pneumoniae*, *S. enteritidis*, *L. innocua*, and *S. aureus* was assumed to be greater than 20, 5, 10, 10, 10 mJ/cm^2^, respectively, in both PBS and TSA media. However, different killing rates were shown at 4.5 mJ/cm^2^ energy density in both conditions, such as 34.2% (PBS) and 49.5% (TSA) (Figure 5a), and 40.1% (PBS) and 22.6% (TSA) (Figure 5d). As shown in Figure 5 (a; *E. coli*) and (d; *L. innocua*), lower energy density resulted in a higher standard deviation of the killing rate due to a sub-optimal level of energy density.

### 3.4. Disinfection in the Solid Surface

The disinfection results on the solid surface (tile surface, red circle) and in nutrient-rich agar media (TSA, black rectangle) are shown in Figure 6. These results were similar for PBS and TSA media; however, results for the tile surface showed a different regime. The killing rates were the same in the case of *K. pneumoniae*; however, the other species showed a higher killing rate in general when tested on tile surfaces rather than TSA media. For example, a 100% killing rate of *E. coli* and *S. aureus* was achieved at 13.5 and 4.5 mJ/cm^2^, respectively (shown in Figure 6a,e). In addition, it could also be seen that the killing rate was higher at 4.5 mJ/cm^2^ energy dose for *E. coli* and *L. innocua*.

The averaged log reduction values of five distinct species in three different conditions are summarized in Table 1 and Table 2. Energy density conditions listed in Table 1 and Table 2 are 4.5 and 9.0 mJ/cm^2^, respectively. The average log reductions values were estimated using triplicate samples. The average initial number of colonies (*N*_o_) was calculated from three different control plates, i.e., three different numbers (*N*_0-1_, *N*_0-2_, *N*_0-3_) were determined from each control plate and averaged. Standard deviation (SD) values are shown in brackets. As shown in both tables, averaged log reduction values were similar for PBS and TSA media; however, higher log reduction values were achieved for the tile surface. From these results, it was demonstrated that disinfection could be achieved in various matrices, and an appropriate energy dose should be required for each species.

### 3.5. Comparison of the Log Reduction Values

A series of survival plots representing five different microorganisms in three different environmental conditions is shown in Figure 7. Log reduction values were obtained by increasing energy densities while treating a fixed initial concentration of 10^3^ CFU/mL. Note that the log reduction value could not be displayed for a 100% killing rate. In both liquid and nutrient-rich agar, the highest energy density was needed to reduce the concentration of *E. coli*. Additionally, the reduction rate for *E. coli* was considerably lower than that of the other organisms. For solid tile surfaces, all tested organisms showed a similar rate of reduction. For example, the averaged slope of *E. coli* in PBS, TSA agar, and tile surface is –0.10, –0.09, and –0.19, respectively.

## 4. Discussion

In this study, simultaneous contamination detection and disinfection were accomplished in various environmental conditions using a portable device. Although the selected device has two different excitation sources (UV and visible light), only the visible wavelength (405 nm) was used as an excitation source in the detection mode, since UV light could inactivate several microorganisms.

In the case of *E. coli* in three different dilutions, it was observed that inoculation concentrations of 4.51 × 10^4^ and 3.18 × 10^3^ CFU/mm^2^ when dispensed on the slide glass or solid tile surface generated a sufficient level of fluorescence signal to be visible in the detection mode. Similarly, S. A. Hice [35] reported that fluorescence intensities of *E. coli* in concentrations greater than 10^3^ CFU/mm^2^ are detectable with a paper-based assay using a low-cost fluorescent reader device. Chen et al. [36] also reported that a fluorescent microscope could capture *Salmonella enterica* in chicken meat in concentrations ~7 × 10^3^ CFU/mm^2^. In another example, Sohn et al. [37] reported that a clear 345 nm fluorescence signal from *E. coli* suspension (below 10^6^ CFU/mL) resulting from two excitation wavelengths at 225 nm and 280 nm can be readily detected.

In addition, the size limit of detectable contaminant was determined to be about 0.15 mm for a single colony on an agar plate. Similarly, Sueker et al. [28] reported that the minimum detectable spot size for saliva and respiratory droplets on stainless steel plates was about 0.13 mm, using a CSI-D device (a previous model of CSI-D+). These limitations, such as detectable concentrations or spot size, are also affected by the measurement parameters, such as working distance and angle of incidence of the excitation light. The fluorescence-based detection approach might be improved by changing the direction of excitation light from orthogonal to a diagonal axis on the target surface. Similarly, several researchers reported that fluorescence emission was detected at the orthogonal angle to the excitation beam to minimize interference [38,39,40].

Disinfection efficacy can be affected by several different design parameters, including wavelength, number of LEDs used, spatial arrangement of the LEDs, and so on. Several preliminary studies are currently under development to optimize uniform irradiation of UVC light; for instance, Kim et al. [41] reported that a linear array of UVC LEDs was installed in a bar-type module. The CSI-D+ consists of one large and one small circular arrays of UVC LEDs to provide a more uniform UV dose on the target surface. This device also included a LIDAR sensor, which provides information for adjusting a specific energy density, since energy density is inversely proportional to the square of the distance from the energy source to the target surface. One of the benefits of the CSI-D+ is that this unit provides real-time target distance and energy density so that inspectors can cross-check with the required energy dose to effectively disinfect certain surfaces. If the distance is larger than the specified length, the user can increase the energy dose to compensate for the loss of energy density.

During the disinfection process, a suitable energy dose is necessary for the effective disinfection of species. Microbial species are categorized based on their cell-wall characteristics as gram-negative or gram-positive microorganisms [42]. Specifically, different level of energy are required to break their chemical bonds, DNA, and cellular membrane (peptidoglycan and/or lipopolysaccharide membrane) since sensitivity to UV is correlated strongly with genome length [30]. In addition, different optical properties of the surrounding area of the target samples can affect the overall inactivation efficiency [43]. In this study, resistance to UVC was found to be in the order *K. pneumoniae* (lowest resistance), *S. aureus*, *S. enteritidis*, *L. innocua*, and *E. coli* (highest resistance). For example, all colonies of *K. pneumoniae* were not viable after exposure to about 5 mJ/cm^2^ in every experiment; however, *E. coli* needed an energy dose of at least 20~25 mJ/cm^2^ for effective killing. A relatively high SD of log reduction was observed in *E. coli* (see Table 1 and Table 2). Both tested energy conditions (4.5 and 9.0 mJ/cm^2^) were significantly less than the effective dose (22.5 mJ/cm^2^). It may be hypothesized that the lethality of less potent doses is also less consistent. These differential susceptibilities to UVC lead to the same results in the other two environments (agar and tile surface). Similarly, Gunter-Ward et al. reported that inactivation UVC doses for a 5-log reduction for *E. coli* were about twice that required for others, such as *L. monocytogenes* and *S. typhimurium* [34].

In the case of concentration differences, it was shown that there were no clear differences in disinfection rate between relatively higher and lower concentrations, as shown in Figure 4 and Appendix A. A uniform UVC dose may result in a uniform disinfection process over a whole target area, inducing similar log reduction values, depending on concentrations. In addition, we expected to observe resistance to UVC based on the gram type since gram-positive bacteria has only a layer of peptidoglycan, while gram-negative bacteria have both peptidoglycan and lipopolysaccharide cell wall. Instead, it was shown that different dose levels of energy should be applied to each bacterial type, and that the type of cell wall was not the only deciding factor for resistance to UVC. Similarly, Kim and Kang demonstrated [44] that two gram-negative and two gram-positive bacteria showed different inactivation rate constants. Therefore, more thorough testing of other microorganisms should be conducted to be broadened and applicable to the practical situation.

Three different environments, liquid, nutrient agar, and solid, were used to analyze the disinfection of microorganisms in this study. PBS as liquid media was widely used in the conventional disinfection studies, and TSA agars were also commonly selected as nutrient media. In addition, ceramic tile, commonly used in kitchens and hospitals, was selected as a solid surface for the evaluation of disinfection rates. It was shown that similar energy doses were required for the disinfection in PBS and TSA media, resulting in similar log reductions, depending on species types. Interestingly, lower energy doses were required on bright white ceramic surfaces while matching the same concentrations and preparation conditions. It was assumed that reflection and scattering of UV on the layer of tile might induce a minor increase of actual effective dose in targets. Xiao et al. [45] reported that ceramic microstructures have more sources of light scattering owing to grain boundaries, pores, and other factors; however, most UVC light may be transmitted or absorbed in PBS and TSA agars. Similarly, Szczawinski et al. [46], showed that photocatalytic films such as TiO_2_ coated on white ceramic wall tiles are responsible for the enhancement of UV germicidal effects. To validate this hypothesis, the same-size ceramic wall tile with dark color was challenged with an additional disinfection test for *E. coli*. The result showed that an averaged log reduction slope of *E. coli* in the black tile surface was −0.10, which is half the slope value of a white tile surface and similar to that of TSA agar.

It was demonstrated that detection and disinfection through a portable unit could be valuable in certain environments, such as the food industry and hospitals. It was also shown that the selection of an appropriate energy dose could be essential for the effective disinfection of bacterial species. The usage of UV could be effective in real-time and in-field analysis; however, the limitations imposed by penetration depth of various surfaces, such as food products or packaging, still requires further testing and validation.

## 5. Conclusions

The efficacy of a portable device in contamination detection and inactivation was validated in this study. Several gram-negative and gram-positive microorganisms were investigated in realistic environmental situations such as liquid, semi-solid, and solid surfaces. The suitable energy densities on each type of microorganism were estimated while comparing the slope of log reduction plots. The tested device can perform over 3 log reduction of *E. coli*, *K. pneumoniae*, *S. enteritidis*, *L. innocua*, and *S. aureus* within 5, 1, 3, 3, and 2 s of UVC exposure, respectively, on both liquid and semi-solid media. In addition, it was observed that the actual effective dose in the target could vary based on surface properties such as brightness or solidity. Assuming that the optics of the fluorescence detection pathway remain fixed, the detection limit of fluorescence emission depends on the accumulated mass of the biological material; thus, a small number of microorganisms cannot be detected effectively with the current system design. To improve the detection capability, it would be beneficial to investigate the detection efficacy as a function of various optical design parameters, such as different excitation wavelengths and illumination angles, and so forth. A new portable device resulting from this redesign could be used in more realistic settings for surface disinfection, such as a food processing plant or a hospital.

## Figures and Tables

**Figure 1 sensors-23-02135-f001:**
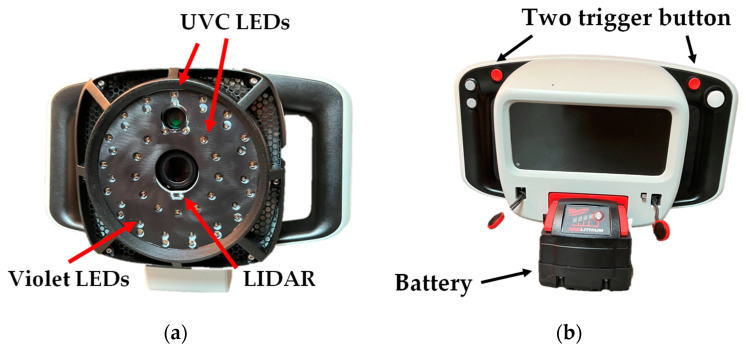
The portable disinfection unit (CSI-D+) used for the present study. (**a**) Without the front cover, showing illumination structure, and (**b**) a frontal view of the unit with a user interface screen.

**Figure 2 sensors-23-02135-f002:**
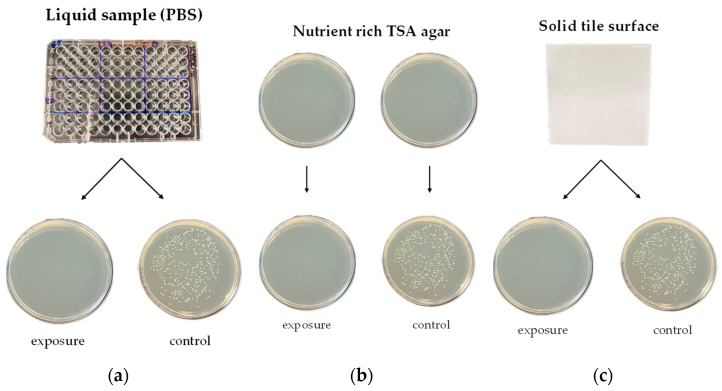
Three different growth environments tested in this study. (**a**) Liquid sample (PBS), (**b**) nutrient-rich TSA agar, and (**c**) solid tile surface.

**Figure 3 sensors-23-02135-f003:**
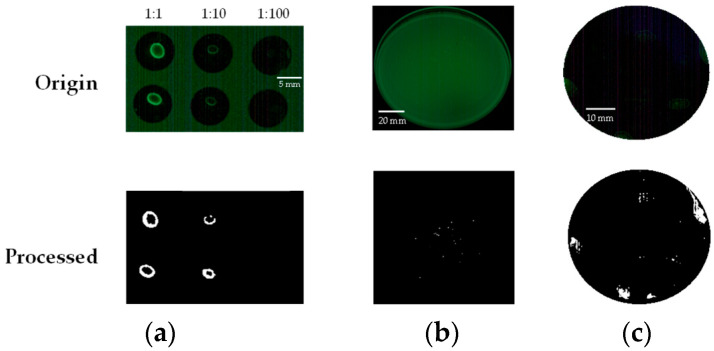
Detection image as (**a**) drop on a slide glass, (**b**) nutrient-rich TSA agar plate, and (**c**) black solid tile surface. The top and bottom pictures represent raw and processed images, respectively.

**Figure 4 sensors-23-02135-f004:**
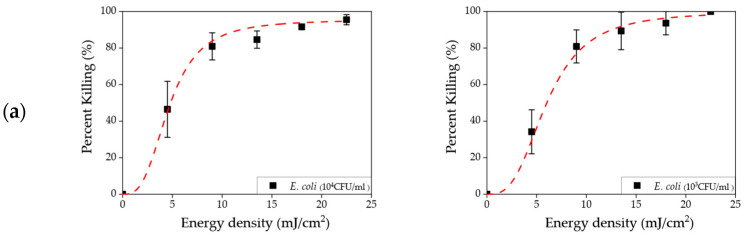
Killing rate as a function of energy density in a liquid sample for (**a**) *E. coli*, (**b**) *K. pneumoniae*, (**c**) *S. enteritidis*, (**d**) *L. innocua*, and (**e**) *S. aureus*. The left and right concentrations are 10^4^ and 10^3^ CFU/mL, respectively.

**Figure 5 sensors-23-02135-f005:**
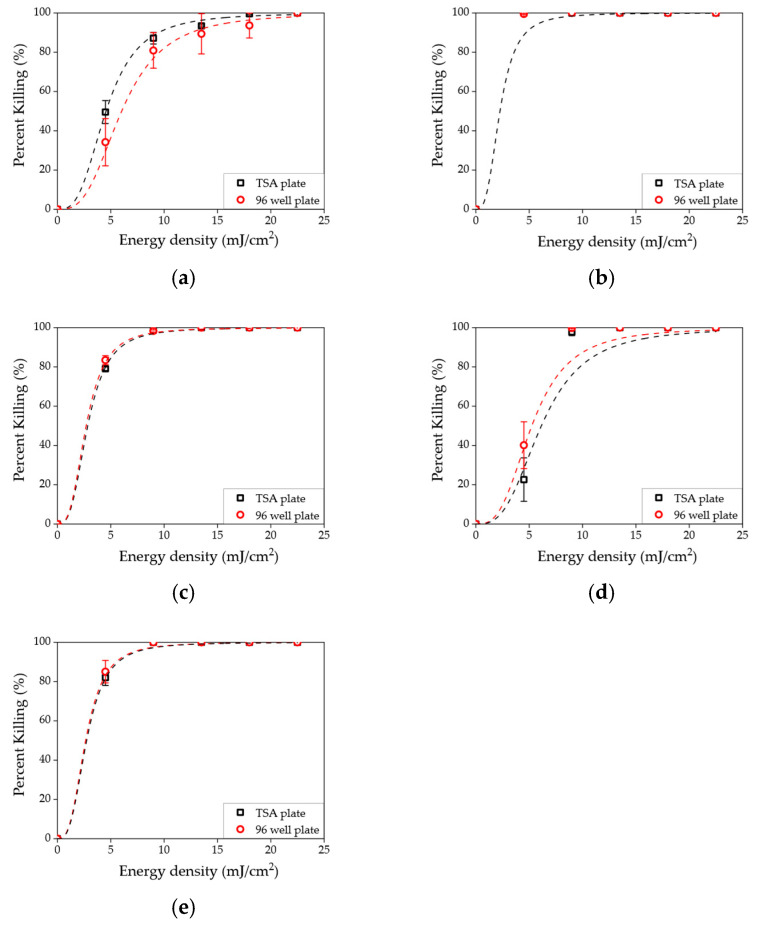
Killing rate as a function of energy density in a liquid sample (black rectangle) and nutrient-rich agar media (red circle) for (**a**) *E. coli*, (**b**) *K. pneumoniae*, (**c**) *S. enteritidis*, (**d**) *L. innocua*, and (**e**) *S. aureus*.

**Figure 6 sensors-23-02135-f006:**
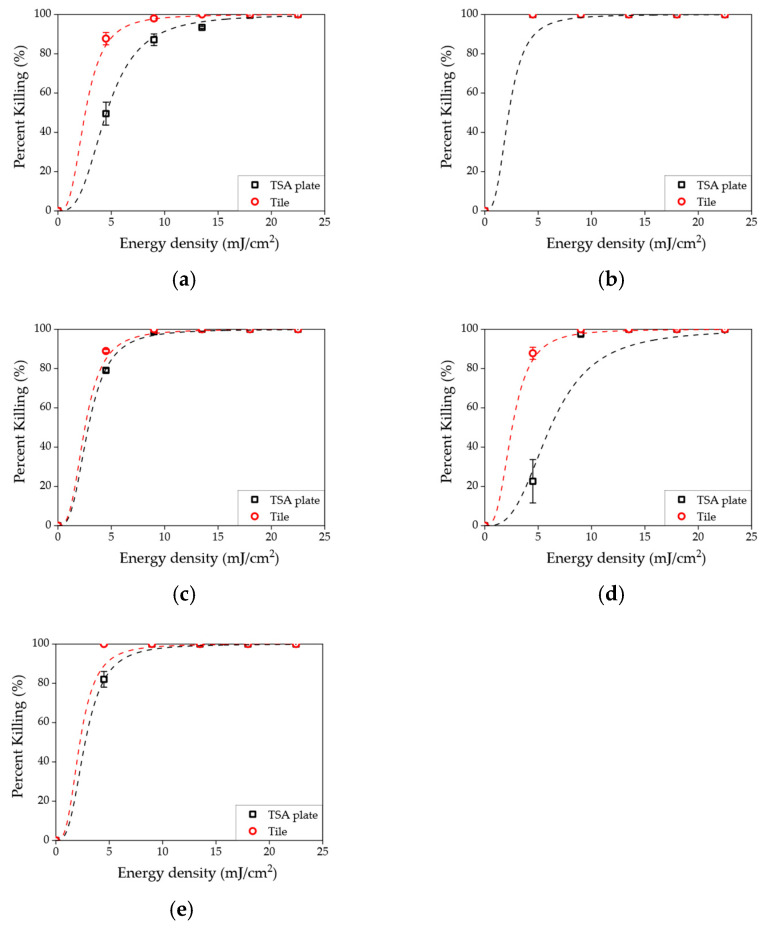
Percentage of killing rate as a function of energy density in nutrient-rich agar media (black rectangle) and solid surface (red circle) for (**a**) *E. coli*, (**b**) *K. pneumoniae*, (**c**) *S. enteritidis*, (**d**) *L. innocua*, and (**e**) *S. aureus*.

**Figure 7 sensors-23-02135-f007:**
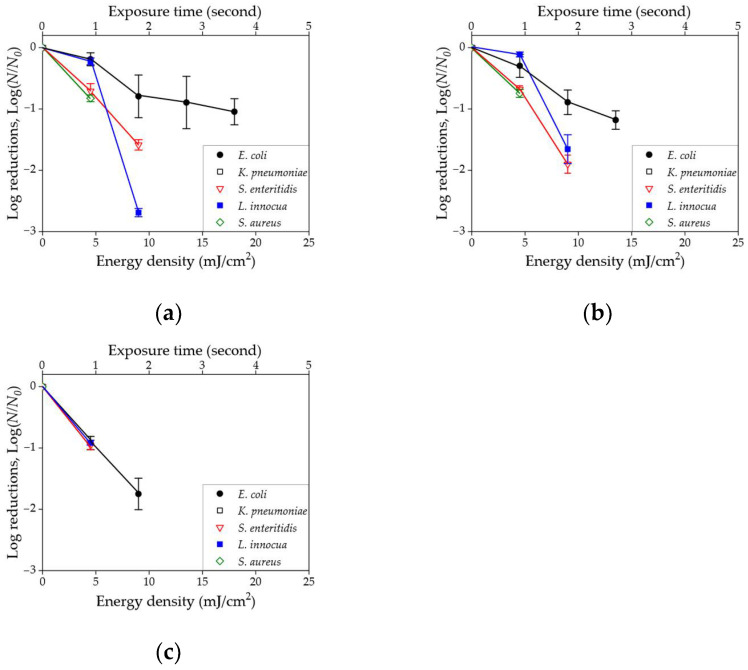
Series of survival plots for five different microorganisms (**a**) in liquid sample, (**b**) on nutrient-rich agar, and (**c**) on solid tile surface.

**Table 1 sensors-23-02135-t001:** Summary table of log reduction values depending on microorganisms at 4.5 mJ/cm^2^ energy density (exposure time of 1 s). Standard deviation values are shown in brackets.

	PBS	TSA Agar	Tile
*E. coli*	–0.190 (0.11)	–0.297 (0.19)	–0.919 (0.10)
*K. pneumoniae*	-	-	-
*S. enteritidis*	–0.708 (0.12)	–0.679 (0.06)	–0.954 (0.07)
*L. innocua*	–0.223 (0.06)	–0.111 (0.04)	–0.914 (0.04)
*S. aureus*	–0.825 (0.06)	–0.746 (0.07)	-

**Table 2 sensors-23-02135-t002:** Summary table of log reduction values depending on microorganisms at 9.0 mJ/cm^2^ energy density (exposure time of 2 s).

	PBS	TSA Agar	Tile
*E. coli*	–0.795 (0.35)	–0.891 (0.20)	–1.749 (0.25)
*K. pneumoniae*	-	-	-
*S. enteritidis*	–1.785 (0.15)	–1.900 (0.15)	-
*L. innocua*	–2.690 (0.07)	–1.656 (0.23)	-
*S. aureus*	-	-	-

## Data Availability

The data presented in this study are available on request from the corresponding author, subject to a confidentiality agreement.

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
