# Peer review of "Surface Environment and Energy Density Effects on the Detection and Disinfection of Microorganisms Using a Portable Instrument"

_sensors, 2023, doi:10.3390/s23042135_

Round 1

Reviewer 1 Report

The manuscript presents a study on an approach to disinfection microorganisms with a portable disinfection unit.

Energy density effects on the disinfection of microorganisms using a portable disinfection unit method are studied.

The authors applied a commercially available device for disinfection and measuring contaminated surfaces. Three types of surfaces (liquid, semi-liquid and solid) contaminated with five species of organisms (3 gram-negative and 2 gram-positive) were considered. Studies were conducted for two different concentrations of the organisms. Surfaces were disinfected with UV-C light and measured using fluorescence spectroscopy with fixed detection mode.

The health safety reason needs to provide fast, practical and possible constant control methods for disinfection of different surfaces. Well-designed studies are constantly necessary to support the promises of commercial devices. The reviewed manuscript is a representation of such studies. It is prepared well enough; the methods and the discussion are consequent. The article can be recommended for publication in Sensors MDPI Journal in its current form. The one revision I suggest is for the Authors to include information on the time of exposure of microorganisms to the UV-C light, i.e. the time of disinfection.

Author Response

We appreciate the comments and critiques from the reviewer. We revised the manuscript following the reviewer’s suggestions. The detailed response to the points raised by the reviewer is attached below.

The one revision I suggest is for the Authors to include information on the time of exposure of microorganisms to the UV-C light, i.e. the time of disinfection.

Following the reviewer’s recommendation, we updated the paper to emphasize the exposure time of disinfection and the correlation between energy density and exposure time was added in 2.4 sections as follows.

“Note that an irradiance of 4.5 mW/cm2 was measured at a fixed working distance, and the energy density was estimated multiplying the irradiance by the exposure time of UVC.”

Therefore, energy density varied from 4.5 to 22.5 mJ/cm2 as well as increasing exposure time from 1 to 5 sec in this study. For example, 9.0 mJ/cm2 (exposure time of 2 sec).

Reviewer 2 Report

The work from Sungho Shin et al reported an efficacy of real-time detection and disinfection of foodborne pathogens, including three gram-negative and two gram-positive microorganisms, of the commercially available instrument, CSI-D+. This work is great in term of the experimental design to validate the detection and disinfection efficiency of the device. In this study, the energy densities, the tested surfaces, the concentrations of the microorganisms have been varied to investigate the performance of the device. The manuscript was clear and well written. The experiments were well planned, and the results were logically discussed. The findings in this manuscript are interesting, significant and will be attracted by broad readers. So, I suggest to accept this manuscript for publication in Sensors.

Minor comments:

1.       The conclusion of this manuscript is too broad and general. The authors have mentioned “Energy density effects” in the title. So, the concise details on energy density effects toward disinfection need to be added.

2.       The statement about “detection of microorganisms” needs to be added in the title to reflect the real application of the device and to match with the journal name (Sensors)    

Author Response

We appreciate the comments and critiques from the reviewer. We revised the manuscript following the reviewer’s suggestions. The detailed response to the points raised by the reviewer is attached below.

1. The conclusion of this manuscript is too broad and general. The authors have mentioned “Energy density effects” in the title. So, the concise details on energy density effects toward disinfection need to be added.

The conclusion in the manuscript has been changed to address the reviewer’s concern. The new version underlines the energy density effects of disinfection.

“The tested device can perform over 3 log reduction of E. coli, K. pneumoniae, S. enteritidis, L. innocua, and S. aureus within 5, 1, 3, 3, and 2 sec of UVC exposure, respectively, on both liquid and semi-solid media. In addition, it was observed that the actual effective dose in the target could vary based on surface properties such as brightness or solidity.”

2. The statement about “detection of microorganisms” needs to be added in the title to reflect the real application of the device and to match with the journal name (Sensors). 

The title has been altered. We added the phrase "microorganism detection." This modification will assist the readers in forming accurate expectations regarding the manuscript's content.

Reviewer 3 Report

the MS is good but needs some revising according to my comments in the pdf.

Author Response

We appreciate the comments and critiques from the reviewer. We revised the manuscript following the reviewer’s suggestions. The detailed response to the points raised by the reviewer is attached below.

1. I think the better modification the tile by adding detection of microorganisms because you detected and disinfection by this device.
The title is changed to “Surface environment and energy density effects on the detection and disinfection of microorganisms using a portable instrument”.

2. In abstract, do not use “we, I, our”. And what is the source of this energy?
The abstract is modified as per the reviewer’s recommendations, and the source of inactivation is updated in an abstract as follows.
“The evaluation of disinfection efficacy was conducted on three distinct types of surfaces, with the energy density of UVC light (275-nm) ranging from 4.5 to 22.5 mJ/cm2 and the exposure time varying from 1 to 5 seconds. The study determined the optimal energy dose for each of the microorganism species”

3. Add references in two equations.
References regarding the calculation of the killing rate and log reduction were added in the manuscript.

4. In results, add statistical analyses. And why SD is high in table 1 and 2.
The description of how the statistical analyses in Tables 1 and 2 were added in the result section 3.4 as follows:
“The average log reduction values were estimated using triplicate samples. The average initial number of colonies (N0) was calculated from three different control plates, i.e., three different numbers (N0-1, N0-2, N0-3) were determined from each control plate and averaged. Standard deviation (SD) values are shown in brackets.”
In this study, E. coli has the highest UVC resistance among the targets. This description was also added in the discussion section as follows.
“A relatively high SD of log reduction was observed in E. coli (See Table 1 and 2). Both tested energy conditions (4.5 and 9.0 mJ/cm2) were significantly less than the effective dose (22.5 mJ/cm2). It may be hypothesized that the lethality of less potent doses is also less consistent."

5. In conclusion, add the limitation of this study, and future prospective.
The conclusion is modified as the reviewer’s recommendations as follows.
“Assuming that the optics of the fluorescence detection pathway remain fixed, the detection limit of fluorescence emission depends on the accumulated mass of the biological material; thus, small number of microorganisms cannot be detected effectively with the current system design. To improve the detection capability, it would be beneficial to investigate the detection efficacy as a function of various optical design parameters, such as different excitation wavelengths and illumination angles, and so forth. A new portable device resulting from this redesign could be used in more realistic settings for surface disinfection, such as a food processing plant or a hospital.”

Reviewer 4 Report

The authors propose the use of a device for contamination detection and inactivation  The authors detailed the study from the choice of the device to the interpretation of the results.  The problem with the article is the novelty.  It is not a novelty in materials and methods. I don't see the point of using a device already tested for the detection of microorganisms. The manufacturer has already done the same thing to see the ability of the device to do this test.  This also gives authors an opportunity to improve the quality of the manuscript.

Author Response

We appreciate the comments and critiques from the reviewer. We revised the manuscript following the reviewer’s suggestions. The detailed response to the points raised by the reviewer is attached below.

The authors propose the use of a device for contamination detection and inactivation. The authors detailed the study from the choice of the device to the interpretation of the results. The problem with the article is the novelty. It is not a novelty in materials and methods. I don't see the point of using a device already tested for the detection of microorganisms. The manufacturer has already done the same thing to see the ability of the device to do this test. This also gives authors an opportunity to improve the quality of the manuscript.

We added the following sentences to clarify the direction and novelty of this work.

“Compared to Sueker et al. [1], this work focuses on expanding the applicability of the proposed instrument in surface contamination detection and disinfection. This was accomplished by preparing samples containing multiple types of microorganisms (a total of five for both gram-negative and gram-positive) and utilizing multiple types of surfaces that may be found in food processing facilities or hospitals.”

In our results, different level of energy densities was required to achieve for the effective killing rate of various species. The tested device can perform over 3 log reduction of E. coli, K. pneumoniae, S. enteritidis, L. innocua, and S. aureus within 5, 1, 3, 3, and 2 sec of UVC exposure, respectively, on both liquid and semi-solid media. For solid surfaces, similar log reductions were obtained with half exposure time. The amount limit of detectable contaminant varied depending on surface types. Inoculations with 4.51×104 and 3.18×103 CFU/mm2, when dispensed on the slide glass and solid tile surface, generated a sufficient level of fluorescence signal for detection.

  1. Sueker, M.; Stromsodt, K.; Gorji, H.T.; Vasefi, F.; Khan, N.; Schmit, T.; Varma, R.; Mackinnon, N.; Sokolov, S.; Akhbardeh, A.; et al. Handheld Multispectral Fluorescence Imaging System to Detect and Disinfect Surface Contamination. Sensors 2021, 21, 7222, doi:10.3390/s21217222.

Round 2

Reviewer 4 Report

The authors have responded to my comments.